# Legionella effector Lpg1137 shuts down ER-mitochondria communication through cleavage of syntaxin 17

Kohei Arasaki[1,*], Yumi Mikami[1,*], Stephanie R. Shames[2], Hiroki Inoue[1], Yuichi Wakana[1] & Mitsuo Tagaya[1]

During infection of macrophages, the pathogenic bacterium Legionella pneumophila secretes effector proteins that induce the conversion of the plasma membrane-derived vacuole into an endoplasmic reticulum (ER)-like replicative vacuole. These ER-like vacuoles are ultimately fused with the ER, where the pathogen replicates. Here we show that the L. pneumophila effector Lpg1137 is a serine protease that targets the mitochondria and their associated membranes. Lpg1137 binds to and cleaves syntaxin 17, a soluble N-ethylmaleimide-sensitive factor attachment protein receptor (SNARE) protein that is known to participate in the regulation of mitochondrial dynamics through interaction with the mitochondrial fission factor Drp1 in fed cells and in autophagy through interaction with Atg14L and other SNAREs in starved cells. Cleavage of syntaxin 17 inhibits not only autophagy but also staurosporine-induced apoptosis occurring in a Bax, Drp1-dependent manner. Thus, L. pneumophila can shut down ER–mitochondria communication through cleavage of syntaxin 17.

[1] Department of Molecular Life Sciences, School of Life Sciences, Tokyo University of Pharmacy and Life Sciences, Tokyo 192-0392, Japan. [2] Department of Microbial Pathogenesis, Yale University School of Medicine, New Haven, Connecticut 06536, USA. * These authors contributed equally to this work. Correspondence and requests for materials should be addressed to K.A. (email: karasaki@toyaku.ac.jp) or to M.T. (email: tagaya@toyaku.ac.jp).

Legionella pneumophila is a Gram-negative intracellular pathogen and is known to cause a serious type of pneumonia called Legionnaire's disease or a less serious one called Pontiac fever. In nature, L. pneumophila replicates inside protozoan hosts such as amoebae, but when inhaled by humans it enters alveolar macrophages and proliferates[1,2]. In macrophages, Legionella secretes effectors through the Dot/Icm apparatus that allow the conversion of the plasma membrane-derived vacuole to the Legionella-containing vacuole (LCV) by modulating membrane transport systems in host cells[3,4]. The LCV is not fused with lysosomes[5] nor undergoes autophagic clearance[6,7], but intimately associates with vesicles derived from the endoplasmic reticulum (ER)[8]. ER-derived vesicles are recruited onto the LCV by the function of Legionella effectors such as DrrA/SidM, a guanine nucleotide exchange factor that activates Rab1[9,10], tethered and fused with the LCV in a soluble N-ethylmaleimide-sensitive factor activating protein receptor (SNARE)-dependent manner[11,12]. As a result, the LCV membrane is converted from a plasma membrane-like structure into an ER/ER–Golgi–intermediate compartment-like structure, which is ultimately fused with the ER to create a compartment that supports bacterial replication[13]. At the initial stage of infection, mitochondria are found to be close to the LCV, whereas at the late stage they decline and ribosomes are observed around Legionella-containing membranes.

Syntaxin 17 (Stx17) is a SNARE originally implicated in a vesicle-trafficking step to the smooth-surfaced tubular ER membranes that are abundant in steroidogenic cells[14]. Stx17 is unique in that it has a long hairpin-like C-terminal hydrophobic domain (CHD), followed by a cytoplasmic basic region. Stx17 participates in cellular events unrelated to membrane fusion. In fed cells, Stx17 promotes mitochondrial fission by defining the localization and activity of the mitochondrial fission factor Drp1 (ref. 15). On starvation, on the other hand, Stx17 dissociates from Drp1 and associates with Atg14L, a subunit of the phosphatidylinositol 3-kinase complex. This promotes the recruitment of this kinase to the mitochondria-associated ER membrane (MAM)[16], which leads to the formation of phosphatidylinositol 3-phophate (PI3P)-enriched omegasomes that are considered to represent a membrane/lipid source for autophagosomes[17,18]. In the late stage of autophagy, Stx17 present on autophagosomes mediates the fusion of autophagosomes with lysosomes[19–21].

In this study, we show that Stx17 is degraded on Legionella infection. We identify the Legionella effector Lpg1137 as the responsible protein for Stx17 breakdown and show that Lpg1137 is a serine protease that localizes to the ER–mitochondria contact site, where Stx17 is located.

## Results

**Stx17 is degraded by the Legionella effector Lpg1137.** In the course of a study to reveal the mechanism underlying Legionella infection and proliferation, we incidentally discovered that this pathogen cleaves Stx17 when engulfed by FcγRII-expressing HeLa cells (Fig. 1a,c) and macrophage-like THP1 cells (Fig. 1d). The cleavage of Stx17 by Legionella must be accomplished by its effectors because no cleavage was observed when cells were infected with an isogenic Legionella dotA mutant defective in effector secretion (Fig. 1c).

To determine which effector is responsible for Stx17 cleavage, we first used isogenic Legionella strains with large chromosomal

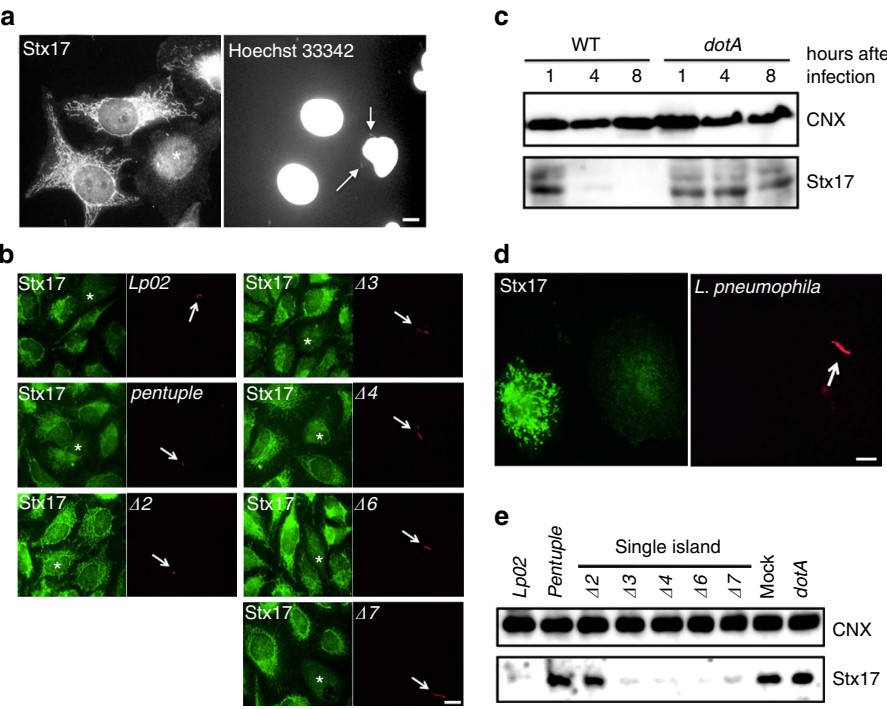

**Figure 1 | Stx17 is degraded on Legionella infection.** (**a**,**b**) HeLa-FcγRII cells were infected with (**a**) wild-type (WT) L. pneumophila or (**b**) one of the indicated strains at an MOI of 5. At 4 h after infection, cells were fixed and double stained with (**a**) an anti-Stx17 antibody and Hoechst 33342, or (**b**) an anti-Legoinella serum. Asterisks and arrows indicate infected cells and Legionella, respectively. Scale bar, 5 μm. (**c**) HeLa-FcγRII cells were infected with the WT Legionella or a dotA mutant strain at an MOI of 50. At the indicated times, cell lysates were prepared, and equal amounts of proteins were subjected to SDS–PAGE followed by IB with antibodies against calnexin (CNX) and Stx17. (**d**) THP1 cells were treated as described in **a** except for infection at an MOI of 25, and double stained with antibodies against Stx17 and Legionella. Scale bar, 5 μm. (**e**) Similar experiments as in **c** except that the indicated strains were used for infection. Uncropped images of blots are shown in Supplementary Fig. 7.

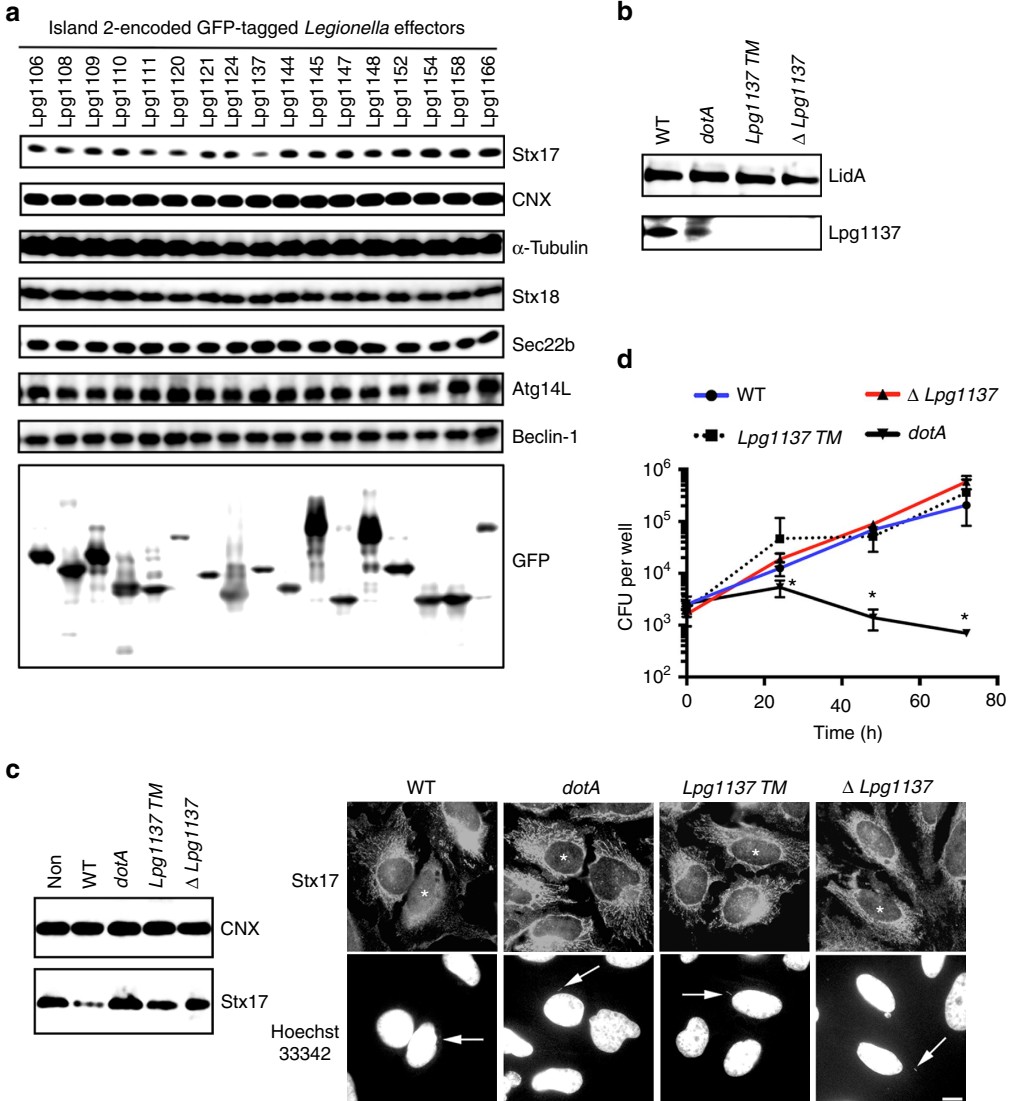

**Figure 2 | Lpg1137 is responsible for the degradation of Stx17. (a)** HeLa-FcγRII cells were transfected with one of the indicated Lpg constructs. At 24 h after transfection, equal amounts of cell lysates were analysed by IB with the indicated antibodies. (**b**) Equal amounts of lysates from the indicated strains of *Legionella* were analysed by IB using antibodies against LidA and Lpg1137. (**c**) HeLa-FcγRII cells were infected with one of the indicated strains of *Legionella* at an MOI of 50 (left) or 5 (right). At 4 h after infection, cells were lysed and analysed by IB with antibodies against CNX and Stx17 (left) or double stained with an anti-Stx17 antibody and Hoechst 33342 (right). Asterisks and arrows indicate infected cells and *Legionella*, respectively. Scale bar, 5 μm. (**d**) Intracellular growth of *Legionella* and mutants. Host cells were infected with wild-type (WT) *Legionella* (blue line), *Lpg1137 TM* (black dot-line), Δ *Lpg1137* (red line) or *dotA* mutant (black solid line). Total colony-forming units (CFU) of *Legionella* were determined at the indicated times as described in Methods. Values are means ± s.d. ($n = 3$). $P < 0.01$ as compared with WT. No significant difference was observed between WT and *Lpg1137 TM* or Δ *Lpg1137* mutant. Uncropped images of blots are shown in Supplementary Fig. 7.

deletions. *L. pneumophila* chromosome has a modular architecture, and five genomic islands (islands 2, 3, 4, 6 and 7) are devoid of genes for growth in rich media, but exhibit a high concentration of effectors that facilitate infection and proliferation in host cells[22]. Stx17 was found not to be cleaved in HeLa-FcγRII cells infected with the *Δ2* mutant lacking 17 effectors, as well as the *pentuple* mutant (Fig. 1b,e). We constructed vectors encoding the individual effectors with a green fluorescent protein (GFP) tag and transfected them into HeLa-FcγRII cells. Stx17 staining was only lost in cells expressing the GFP version of the effector encoded by *lpg1137* (Supplementary Fig. 1). This was confirmed by immunoblotting (IB) (Fig. 2a). Notably, a Stx17-binding SNARE protein, Sec22b[14], another ER SNARE, Stx18 (ref. 23), or an autophagy partner, Atg14L[16,21], was not cleaved, demonstrating the specificity of

Stx17 cleavage by Lpg1137. To confirm that Lpg1137 is responsible for Stx17 cleavage, we constructed a transposon mutant strain (*Lpg1137 TM*) and a deletion mutant strain (Δ*Lpg1137*) (Fig. 2b). Stx17 was not cleaved on infection with these mutants, as in the case of the *dotA* mutant (Fig. 2c). In contrast to the *dot*A mutant, on the other hand, these mutants did not exhibit a growth defect (Fig. 2d).

**Lpg1137 is a serine protease localized in the MAM/mitochondria.** To characterize Lpg1137, we first investigated its localization by subcellular fractionation. In addition to the cytosolic and microsomal fractions, GFP-Lpg1137 was recovered in the MAM and mitochondria fractions, where Stx17 resides (Fig. 3a). We therefore reasoned that Lpg1137 targets the MAM and mitochondria, and directly binds to and cleaves Stx17. To test this

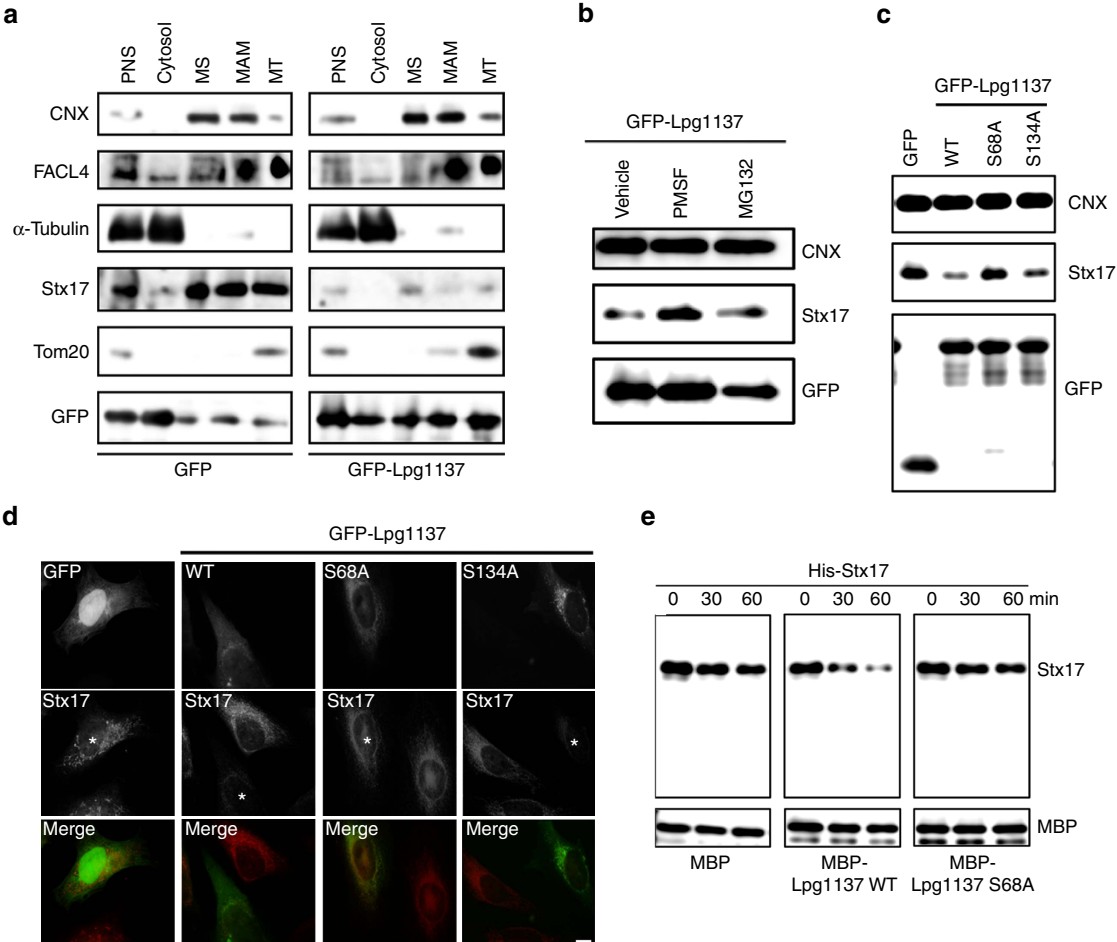

**Figure 3 | Lpg1137 is a serine protease localized in MAM/mitochondria.** (**a**) HeLa-FcγRII cells were transfected with a plasmid encoding GFP (left) or GFP-Lpg1137 (right). At 24 h after transfection, cells were subjected to subcellular fractionation, and equal amounts of fractions were analysed by IB with the indicated antibodies. MS and MT denote microsomes and mitochondria, respectively. (**b**) HeLa-FcγRII cells were transfected with GFP-Lpg1137. At 4 h after transfection, DMSO (Vehicle), PMSF (1 mM) or MG132 (1 μM) was added to cells, and the cells were incubated for 20 h. Equal amounts of cell lysates were analysed by IB with the indicated antibodies. (**c,d**) HeLa-FcγRII cells were transfected with one of the indicated plasmids, and after 24 h equal amounts of lysates were analysed by IB with the (**c**) indicated antibodies. Alternatively, cells were fixed and stained with (**d**) an anti-Stx17 antibody. Scale bar, 5 μm. (**e**) His-Stx17 (0.2 μg) was incubated with MBP, MBP-Lpg1137 wild-type (WT) or MBP-Lpg1137 S68A (each 0.2 μg) for the indicated times at 37 °C. After incubation, samples were subjected to IB with antibodies against Stx17 and MBP. Uncropped images of blots are shown in Supplementary Fig. 7.

idea, we transfected cells with a vector encoding GFP-Lpg1137 and then treated them with a serine protease inhibitor, phenylmethylsulfonyl fluoride (PMSF), or a proteasome inhibitor, MG132. We found that PMSF, but not MG132, substantially blocked Stx17 cleavage (Fig. 3b). Sequence analysis of Lpg1137 showed that Ser68 and the surrounding residues (-Gly-Leu-Ser-Gly-Gly-) fit the consensus sequence for the active site of serine proteases (Gly-$X$-Ser-$X$-Gly/Ala, where $X$ is any residue). Residues 132–136 (-Gly-Leu-Ser-Gly-Lys-) exhibit a similar sequence. We individually replaced Ser68 and Ser134 with Ala (the S68A and S134A mutants, respectively) and then the mutants proteolytic activities towards Stx17 were investigated. Stx17 was not cleaved by the S68A mutant (Fig. 3c,d), whereas S134A cleaved Stx17, perhaps slightly less efficiently than wild-type Lpg1137 (Fig. 3c,d). Finally, we directly tested the proteolytic activity of Lpg1137 using recombinant proteins. Recombinant His$_6$-tagged Stx17 was cleaved by the maltose-binding protein (MBP)-tagged Lpg1137 wild-type but not by the S68A mutant (Fig. 3e).

As the S68A mutant lacks proteolytic activity, we could define the binding site for Lpg1137 on Stx17. The GFP-tagged S68A

mutant bound to FLAG-Stx17, but barely to FLAG-Stx18 (Fig. 4a). Stx17 has a hairpin-like C-terminal hydrophobic domain comprising 44-amino-acid residues followed by the cytoplasmic C-terminal region (CHD + C) that are required and sufficient for Stx17 to localize to the MAM/mitochondria and to interact with the mitochondrial fission factor Drp1 (ref. 15). Replacement of Lys254 with other amino acids except for Arg causes redistribution of Stx17 to the entire ER with loss of the Drp1-binding ability[15]. Analysis involving truncation mutants of Stx17 showed that the CHD + C region, not the SNARE motif, is the region interacting with Lpg1137 (Fig. 4b). In addition, as in the case of the binding to Drp1, the K254C mutant exhibited reduced binding to Lpg1137, which is consistent with the fact that the K254C mutant was not cleaved on infection with *Legionella* (Fig. 4c).

**Legionella eliminates the function of Stx17.** We examined whether *Legionella* blocks the function of Stx17. The fact that Stx17 is cleaved by Lpg1137 predicts that the interactions of Stx17 with Atg14L in starved cells and Drp1 in fed cells are eliminated on *Legionella* infection. We investigated this prediction using

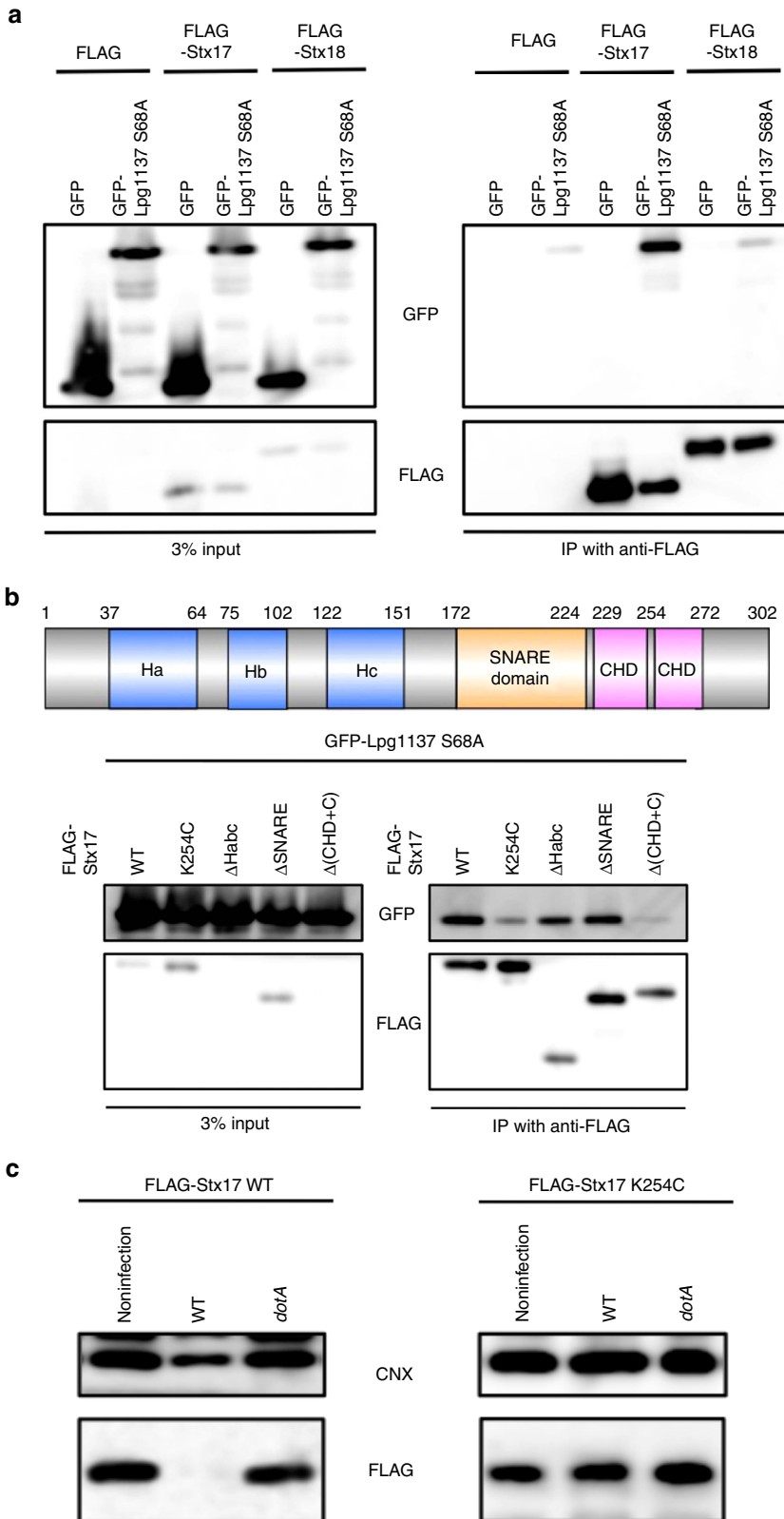

**Figure 4 | Stx17 interacts with Lpg1137 through the CHD + C region.** (**a**) HEK293-FcγRII cells were co-transfected with plasmids encoding GFP-Lpg1137 S68A and either FLAG, FLAG-Stx17 or FLAG-Stx18. At 24 h after transfection, cell lysates were prepared and immunoprecipitated with FLAG-M2 beads. The precipitated proteins were analysed by IB with antibodies against GFP and FLAG. (**b**) Schematic representation of Stx17 (upper panel). HEK293-FcγRII cells were co-transfected with plasmids encoding GFP-Lpg1137 S68A and FLAG-tagged full-length Stx17 or one of its mutants. Cell lysates were immunoprecipitated and analysed by IB with antibodies against GFP and FLAG (lower panel). (**c**) HeLa cells stably expressing FLAG-tagged wild-type (WT) Stx17 or the K254C mutant were transfected with a plasmid encoding non-tagged FcγRII. At 24 h after transfection, cells were non-infected or infected with WT *Legionella* or a *dotA* mutant at an MOI of 50 for 4 h. Equal amounts of cell lysates were analysed by IB with antibodies against CNX and FLAG. Uncropped images of blots are shown in Supplementary Fig. 7.

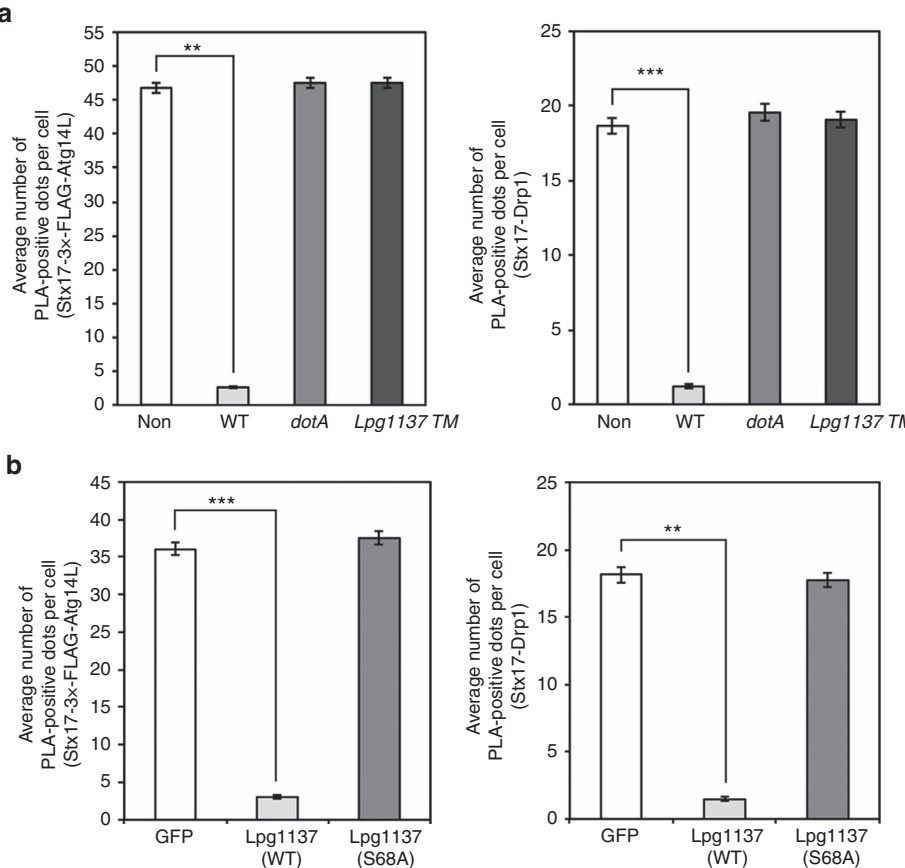

**Figure 5 | Lpg1137 eliminates the interactions of Stx17 with Atg14L and Drp1. (a)** HeLa-FcγRII cells expressing 3 × -FLAG-Atg14L (left) or HeLa-FcγRII cells (right) were non-infected or infected with *Legionella* at an MOI of 50 for 4 h. Before fixation, HeLa-FcγRII cells expressing 3 × -FLAG-Atg14L were incubated with EBSS for 2 h to induce autophagy. Cells were fixed and subjected to PLA using the indicated pairs, and the number of PLA dots was scored. Values are means ± s.e.m ($n=3$). **$P<0.001$ and ***$P<0.0001$ as compared with noninfection. **(b)** HeLa-FcγRII cells were transfected with a plasmid encoding GFP, GFP-Lpg1137 wild-type (WT) or GFP-Lpg1137 S68A with or without 3 × -FLAG-Atg14L. At 24 h after transfection, cells expressing GFP constructs and 3 × -FLAG-Atg14L were incubated with EBSS for 2 h and fixed, or cells expressing only GFP constructs were fixed. PLA was performed using the indicated pairs. Values are means ± s.e.m. ($n=3$). **$P<0.001$ and ***$P<0.0001$ as compared with GFP.

proximity ligation assay (PLA)[24]. When wild-type *Legionella* was used at a multiplicity of infection (MOI) of 50, almost all HeLa-FcγRII cells were infected (data not shown). Under this condition, PLA dots representing the proximity of Stx17 to 3 × -FLAG-Atg14L in starved cells and to Drp1 in fed cells were largely diminished, whereas no significant decrease in the PLA signals was observed in cells infected with the *dotA* or *Lpg1137 TM* mutant (Fig. 5a and Supplementary Fig. 2a). Moreover, the PLA signals for Stx17-3 × -FLAG-Atg14L and Stx17-Drp1 were abolished on expression of GFP-Lpg1137 wild-type, but not GFP-Lpg1137 S68A (Fig. 5b and Supplementary Fig. 2b).

**Expression of Lpg1137 blocks autophagy.** Given that Lpg1137 eliminated the Stx17-Atg14L interaction by cleaving Stx17, we next examined whether Lpg1137 could block autophagy. When cells expressing the Lpg1137 wild-type were starved for 2 h in the absence or presence of bafilomycin A1, the appearance of the autophagosome marker LC3 was reduced compared with in control cells (Fig. 6a). In contrast, no change in the formation of LC3 puncta was observed in cells expressing the S68A mutant (Fig. 6a). Inhibition of autophagy by Lpg1137 was confirmed by the reduced accumulation of LC3-II observed on IB (Fig. 6b).

Autophagy starts with the formation of ER-connected omegasomes marked by the PI3P effector DFCP1 (refs 25,26). A previous study demonstrated that Stx17 recruits the PI3-kinase

complex to the MAM through interaction with Atg14L[16], a subunit of the kinase complex[27]. We therefore examined the localization of FLAG-DFCP1 and 3 × -FLAG-Atg14L in GFP-Lgp1137-expessing cells after starvation. In GFP-expressing cells (control), both FLAG-DFCP1 and 3 × -FLAG-Atg14L exhibited punctate staining, suggesting the formation of PI3P-positive omegasomes (Fig. 6c). In contrast, no discrete puncta for FLAG-DFCP1 or 3 × -FLAG-Atg14L was observed in cells expressing the GFP-Lpg1137 wild-type. Expression of GFP-Lpg1137 S68A did not affect the formation of puncta for FLAG-DFCP1 and 3 × -FLAG-Atg14L. These results suggest that GFP-Lpg1137 blocks an early stage of autophagy, that is, PI3P formation on omegasomes. Similar results were obtained when Stx17 was knocked down by short interfering RNA (siRNA) (Fig. 6d), corroborating that Stx17 participates in an early stage of autophagy. To further substantiate this view, we overexpressed RavZ, which inhibits autophagy by cleaving LC3 from preautophagosomal structures[6,7]. As reported, no LC3-positive structures were formed in GFP-RavZ-expressing cells, and the PI3P-binding protein GFP-RavZ was colocalized with mRFP-DFCP1 (Fig. 7a). Notably, FLAG-Stx17 was associated with GFP-RavZ-positive preautophagosomal structures. Moreover, depletion of Stx17 substantially suppressed the accumulation of GFP-RavZ as well as that of mRFP-DFCP1 on putative preautophagosomal membrane structures (Fig. 7b), consistent

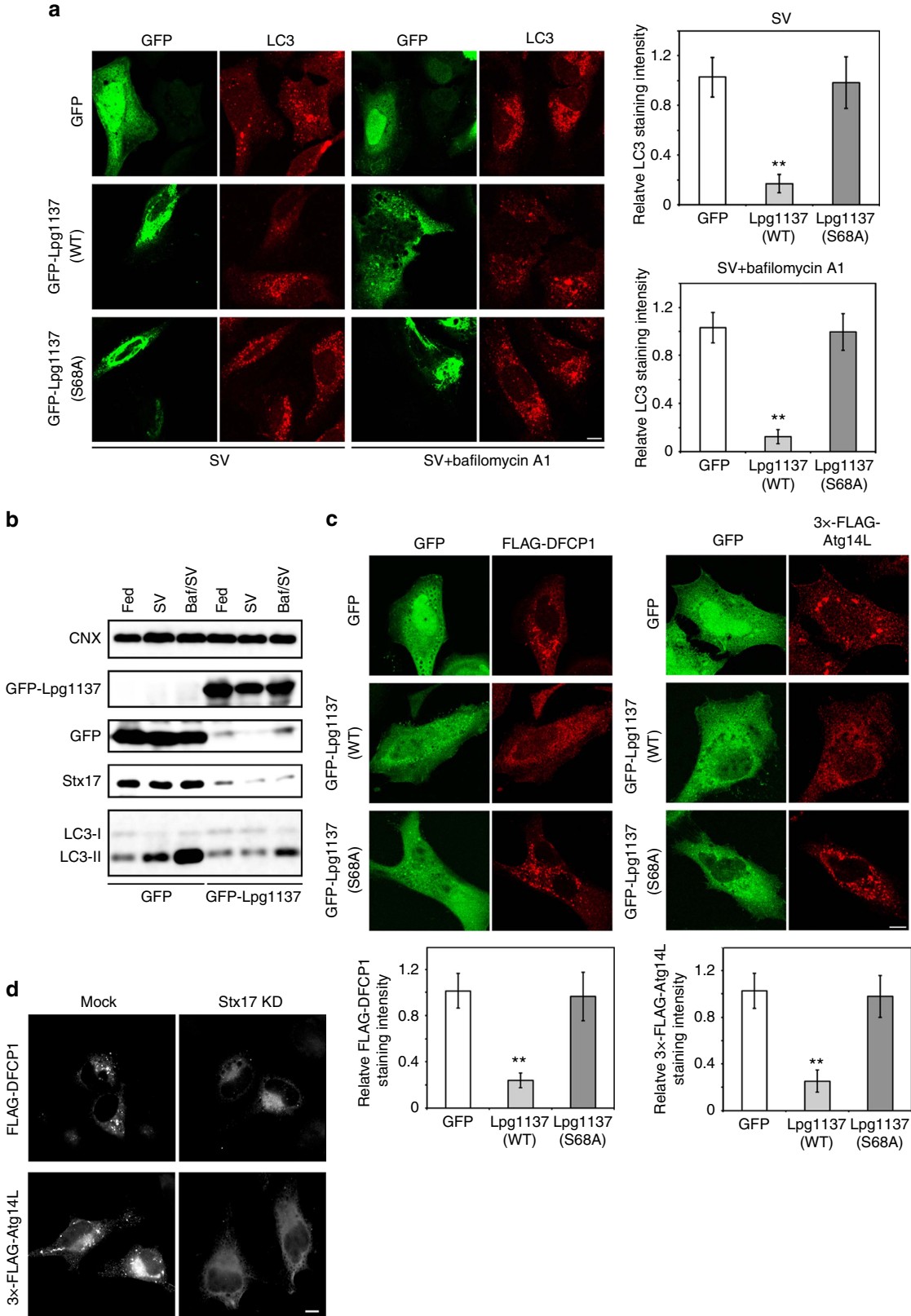

with the view that Stx17 regulates PI3P formation via recruiting the PI3-kinase complex[16].

**Legionella suppresses staurosporine-induced apoptosis.** We explored the possibility that Stx17 cleavage might subvert cellular defense systems other than autophagy. Drp1, a dynamin-like GTPase whose activity is regulated by Stx17 (ref. 15), contributes to apoptosis by facilitating the recruitment of proapoptotic protein Bax to mitochondrial constriction sites[28,29]. In addition, a recent study showed that mitochondrial-anchored protein ligase

(MAPL)-mediated SUMOylation of Drp1 stabilizes the MAM/ mitochondria platform for apoptosis[30]. We therefore examined the effect of GFP-Lpg1137 expression on apoptosis induced by the kinase inhibitor staurosporine (STS) and TNF-related apoptosis-inducing ligand ( TRAIL), which preferentially activate the proapoptotic proteins Bax and Bak, respectively[31]. In cells expressing the GFP-Lpg1137 wild-type, the degrees of nuclear condensation, caspase-3 activation and cytochrome c release from mitochondria induced by STS were markedly lower than those in GFP- or GFP-Lpg1137 S68A-expressing cells, whereas no difference was observed when apoptosis was induced by TRAIL (Fig. 8a and Supplementary Fig. 3). Similar inhibition of STS-induced apoptosis was observed in cells infected by Legionella, but not by the dotA or Lpg1137 TM (Supplementary Fig. 4), and in cells depleted of Stx17 by siRNA (Supplementary Fig. 5). Drp1 knockdown also blocked apoptosis induced by STS, but not by TRAIL (Supplementary Fig. 5). Notably, Bax failed to be translocated to mitochondria in cells depleted of Stx17 or Drp1 (Fig. 8b and Supplementary Fig. 6), and therefore did not accumulate at mitochondrial constriction or fission sites (Fig. 8c) or at Drp1-positive structures (Fig. 8d) in cells depleted of Stx17.

## Discussion

In the present study, we showed that Legionella Lpg1137 is a serine protease that targets the MAM/mitochondria, where it binds to and cleaves Stx17. No cleavage was observed for Sec22b or Atg14L, which are Stx17-binding partners in fed and starved cells[14–16], respectively, suggesting the specificity of the Lpg1137 proteolytic activity. Cleavage of Stx17 blocks an early step of autophagy and STS-induced apoptosis. These effects were also observed in cells depleted of Stx17 by siRNA.

Studies on the function of Stx17 in autophagy showed, somewhat in contradiction, that this protein participates in both an early stage (autophagosome biogenesis)[16] and a late stage (autophagosome–lysosome fusion)[19–21]. The former is in line with the fact that mitochondria supply autophagosomes with lipids and proteins[32]. However, the interaction of Stx17 with Atg14L does not necessarily imply the involvement of Stx17 in the early stage of autophagy because Atg14L functions not only as a subunit of the PI3-kinase complex[27] but also controls Stx17-mediated autophagosome fusion with lysosomes[21]. In the present study, using RavZ, a protease that cleaves the bond between LC3 and phosphatidylethanolamine[6,7], we showed that Stx17 accumulates at areas containing PI3P in the absence of LC3. Moreover, depletion of Stx17 by overexpression of Lpg1137 or siRNA substantially reduced the formation of PI3P, as visualized by the absence of DFCP1-positive puncta. Thus, our data suggest that Stx17 participates in an early stage of autophagy. Lack of inhibition of autophagosome formation by silencing Stx17 observed in a previous study[19] might reflect the difference in Stx17 dependency of the early and late autophagy steps;

autophagosome formation might be less sensitive to Stx17 depletion than autophagosome–lysosome fusion. The degree of Stx17 depletion in a previous study[19] might not have been enough for the inhibition of PI3-kinase complex recruitment step, but enough for the blockage of the fusion step.

In this study, we revealed a mechanism for the blockage of apoptosis by Legionella; cleavage of Stx17 by Lpg1137. For continued intracellular replication, Legionella needs to ensure the survival of the host cell against stresses by toxic microbial products, massive protein synthesis and the immune system. Legionella appears to use multiple mechanisms to prevent apoptosis. On infection of cells, the expression of antiapoptotic proteins is enhanced in host cells[33,34]. The Legionella effector SidF antagonizes proapoptotic proteins[35]. In addition, SdhA prevents apoptosis by an unknown mechanism[36]. A recent study reported that Legionella effectors such as Lgt1 and Lgt2 inhibit unfolded protein response by blocking the splicing of XBP1u mRNA, thus preventing host cells from unfolded protein response-driven apoptosis[37].

In conclusion, the present observations in conjunction with the previous finding that Stx17 depletion impairs ER-mitochondrial $Ca^{2+}$ homeostasis[15] suggest that Legionella can shut down the ER–mitochondria communication by cleaving Stx17. Given that the MAM and Stx17 are involved in a variety of cellular signalling and membrane trafficking pathways[14–16,19,38], including the removal of oxidized or damaged mitochondrial content via mitochondria-derived vesicles[39], it is tempting to speculate that cellular systems other than autophagy and apoptosis might be also subverted on Legionella infection.

In the course of this study, it was reported that the Legionella effector sphingosine-1 phosphate lyase (LpSpl) interferes with autophagy by affecting host sphingolipid metabolism[40]. Future work is required to reveal the interplay among multiple effectors (RavZ, Lpg1137 and LpSpl) to suppress autophagy, as well as the strategic advantage for the presence of the multiple autophagy suppressors in infection and intracellular growth of Legionella.

## Methods

**Animals.** All animal procedures and experiments were approved by the Animal Care Committee of Tokyo University of Pharmacy and Life Sciences and conducted according to the guidelines of the committee.

**Bacterial strains and plasmids.** The conditions for the growth of L. pneumophila strains (the wild-type (Lp01), a thymidine auxotroph mutant (Lp02), a dotA mutant, a pentuple mutant and single island deletion mutants) and host cell infection were as described previously[6,22,41–43]. The lpg1137::tn strain is in background SRS43 (Lp02::thyA (made by allelic exchange to incorporate the wild-type thyA gene)). The transposon mutants were made by electroporation of the plasmid pSRS_Cm1, which is a derivative of the pSAM_bt[44] with the original Erm resistance swapped for chloramphenicol resistance. In-frame deletion of the Lpg1137 gene was performed in Lp02 using allelic exchange, as described previously[43]. A cDNA fragment of Lpg1137 was generated by polymerase chain reaction (PCR) using primer pairs 1137 knockout (KO)-5F (5′-ATTGAGCTCTCC GAATCTTCATAATCTGG-3′) and 1137KO-5R (5′-ATTGCGGCCGCTTGAAT

**Figure 6 | Ectopic expression of Lpg1137 blocks autophagosome formation.** (**a**) HeLa-FcγRII cells were transfected with a plasmid encoding GFP, GFP-Lpg1137 wild-type (WT) or GFP-Lpg1137 S68A. At 24 h after transfection, cells were incubated with EBSS (SV: left two columns) or EBSS plus 0.1 μM bafilomycin A1 (SV + bafilomycin A1, right two columns) for 2 h. After incubation, cells were fixed and stained with an anti-LC3 antibody. The bar graphs on the right show the ratio of the LC3 staining intensity of GFP-expressing to that of -nonexpressing cells. Values are means ± s.d. (n = 3). **P < 0.001 as compared with GFP-Lgp1137 S68A. Scale bar, 5 μm. (**b**) HeLa-FcγRII cells were transfected with a plasmid encoding GFP or GFP-Lpg1137. At 24 h after transfection, cells were incubated with EBSS (SV) ± 0.1 μM bafilomycin A1 (Baf) for 2 h and then lysed. Equal amounts of the lysates were analysed by IB with the indicated antibodies. (**c**) HeLa-FcγRII cells expressing FLAG-DFCP1 (left two columns) or 3×-FLAG-Atg14L (right two columns) were transfected with a plasmid encoding GFP, GFP-Lpg1137 WT or GFP-Lpg1137 S68A. At 24 h after transfection, cells were incubated with EBSS for 2 h, fixed and then stained with an anti-FALG antibody. The bar graphs shown at the bottom exhibit the ratio of the FLAG-DFCP1 or 3×-FLAG-Atg14L staining intensity of GFP-expressing to that of -nonexpressing cells (50 cells each). Values are means ± s.d. (n = 3). **P < 0.001 as compared with GFP-Lpg1137 S68A. Scale bar, 5 μm. (**d**) HeLa-FcγRII cells were transfected without (Mock) or with siRNA targeting Stx17 (Stx17 knockdown (KD)). At 48 h after transfection, cells were transfected with a plasmid encoding FLAG-DFCP1 or 3×-FLAG-Atg14L, incubated for 24 h and then with EBSS for 2 h. The cells were fixed and stained with an anti-FLAG antibody. Scale bar, 5 μm. Uncropped images of blots are shown in Supplementary Fig. 7.

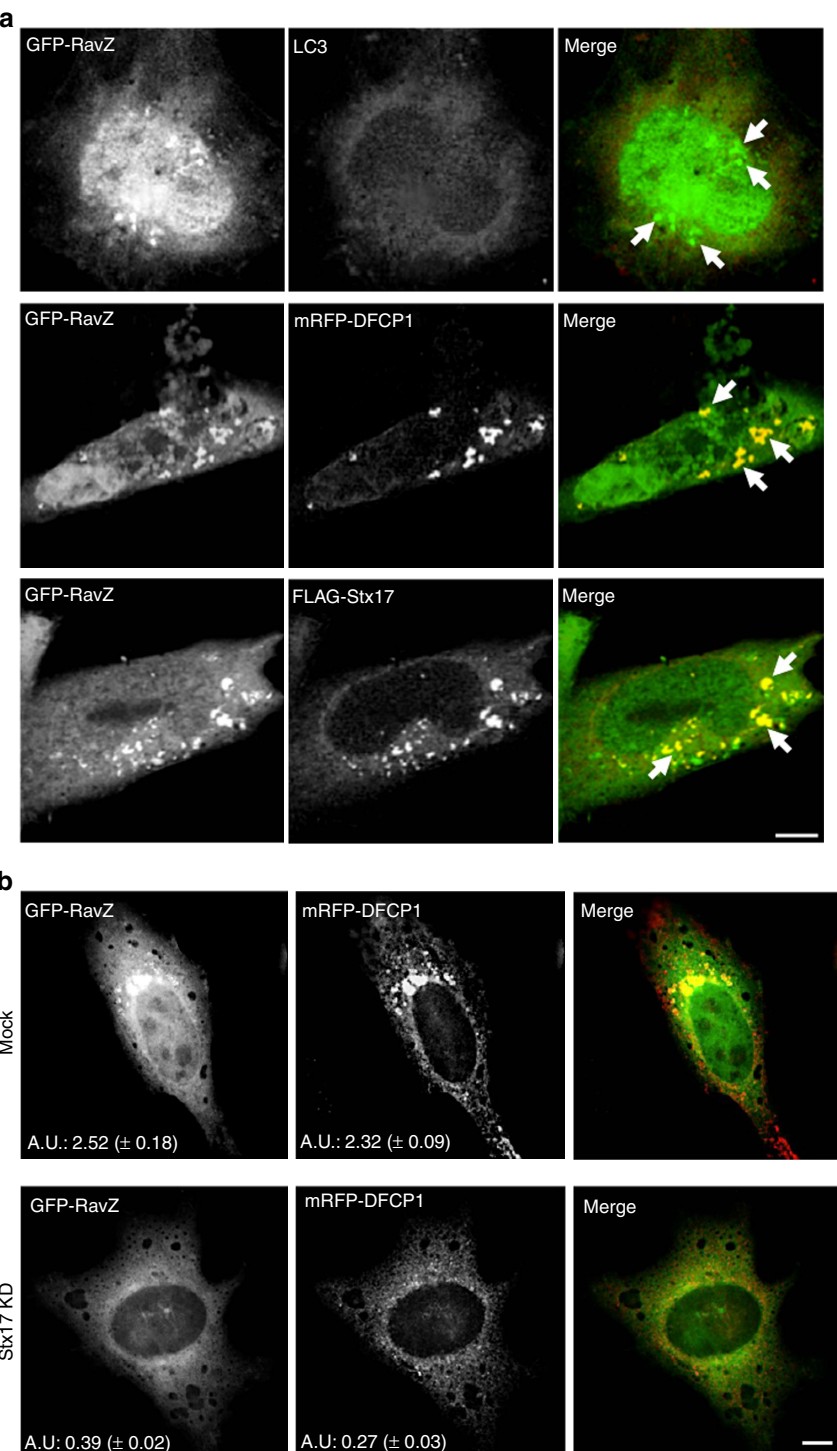

**Figure 7 | RavZ-positive puncta are not formed on Stx17 depletion.** (**a**) HeLa cells stably expressing Stx17 were transfected with a plasmid encoding GFP-RavZ (top and bottom rows) or both GFP-RavZ and mRFP-DFCP1 (middle row). At 24 h after transfection, cells were incubated with EBSS for 2 h, fixed and then stained with an antibody against LC3 (top low) or FLAG (bottom row). Arrows indicate representative GFP-RavZ-positive puncta. Scale bar, 5 µm. (**b**) HeLa-FcγRII cells were transfected without (Mock) or with siRNA (Stx17 knockdown (KD)) targeting Stx17. At 48 h after transfection, cells were transfected with plasmids encoding GFP-RavZ and mRFP-DFCP1. At 24 h after transfection, cells were incubated with EBSS for 2 h, fixed and observed by fluorescence microscopy. The numbers shown at the bottom of each panel indicate the intensity of GFP-RavZ or mRFP-DFCP1 staining in arbitrary unit (a.u.). Fifty cells expressing both GFP and mRFP were analysed in each experiment. Values are means ± s.d. ($n = 3$). $P < 0.01$ (GFP-RavZ) or $P < 0.001$ (mRFP-DFCP1). Scale bar, 5 µm.

CACTTAAATCAATGTGC-3′), and 1137KO-3F (5′-ATTGCGGCCGCAAGAAA TAATCAAGAGTAACAAGG-3′) and 1137KO-3R (5′-ATTGTCGACCAACAA CTGAGTGAAACGCAG-3′), and ligated into the gene replacement vector,

pSR47S[45]. Plasmid encoding the deletion alleles was mated into Lp02, and bacteria in which the plasmid had cointegrated onto the chromosome were selected on CYE agar plates containing kanamycin. PCR analysis was used to identify clones in

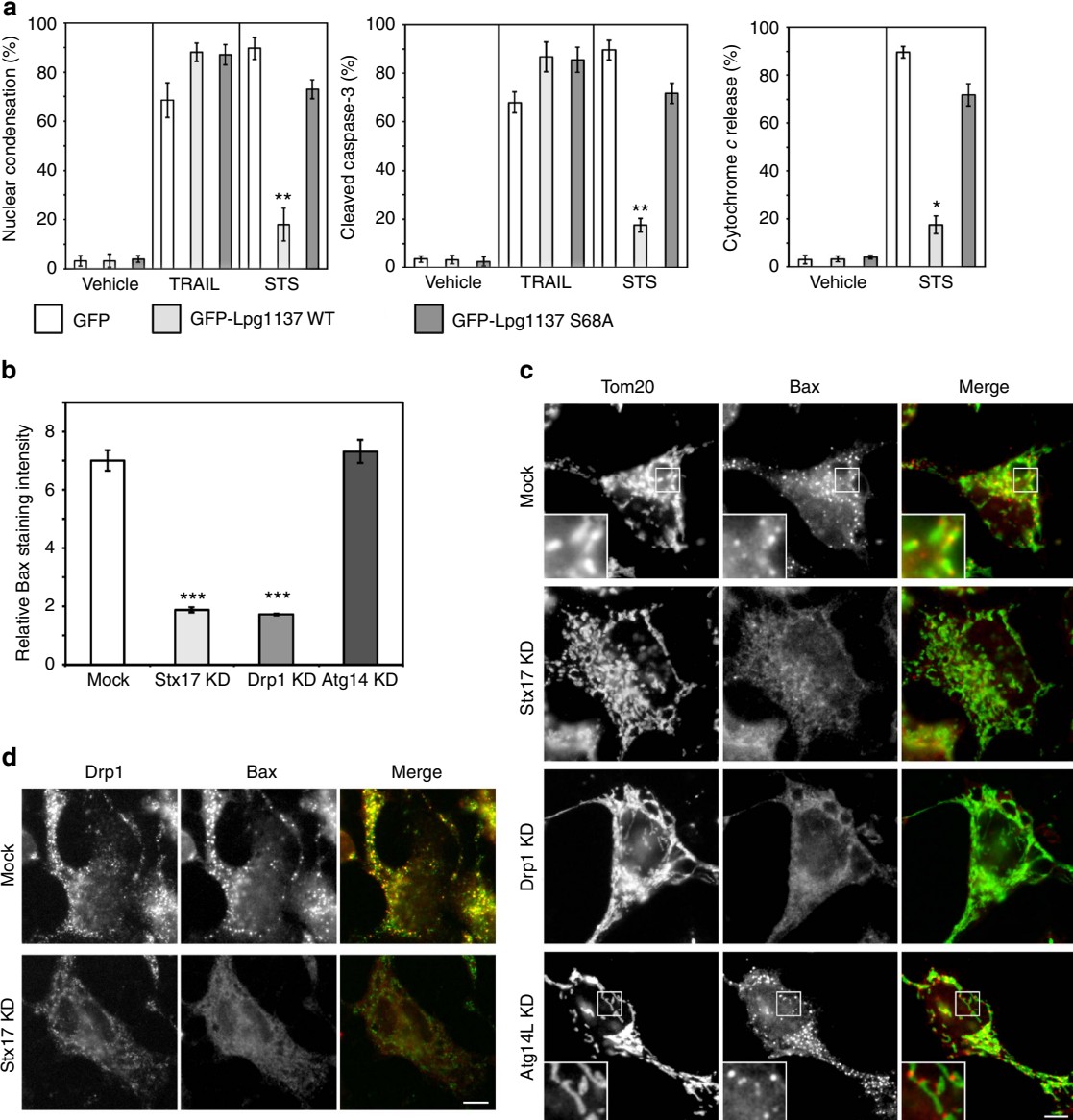

**Figure 8 | Loss of Stx17 suppresses STS-induced apoptosis.** (**a**) HeLa-FcγRII cells were transfected with one of the indicated plasmids. At 24 h after transfection, cells were mock-treated (Vehicle), or treated with 500 ng ml$^{-1}$ TRAIL or 1 μM STS for 4 h and then fixed. Cells exhibiting nuclear condensation (left), cleaved caspase-3 (middle) and cytochrome *c* release from the mitochondria (right) were counted (100 GFP-expressing cells each). Values are means ± s.d. (*n* = 4). *P < 0.01 and **P < 0.001 as compared with GFP-Lpg1137 S68A. (**b**) HeLa-FcγRII cells with mock treatment or depleted (KD) of Stx17, Drp1 or Atg14L were treated with 1 μM STS for 4 h, fixed and then stained with an antibody against Bax (Supplementary Fig. 6). Notably, translocation of Bax to mitochondria increased the staining intensity in cells (Supplementary Fig. 6). One hundred cells were analysed in each experiment. The ratio of the Bax staining intensity between cells after and before STS treatment is plotted. Values are means ± s.d. (*n* = 4). ***P < 0.0001 as compared with Mock. (**c,d**) HeLa-FcγRII cells were treated as described in **b** except for 2.5-h STS treatment to observe an early stage of apoptosis. Cells were fixed and double stained with the indicated antibodies. Scale bar, 5 μm. Insets in **c** are magnifications of boxed areas.

which the deletion allele had replaced the wild-type allele on the chromosome, and loss of Lpg1137 was confirmed by IB using an antiserum against Lpg1137. Plasmid pcDNA-DEST-53 (Thermo Fisher Scientific) was used for the construction of all GFP fusion effectors encoded by the genes in island 2. Substitution of Ser-68 or Ser-134 was performed by inverse PCR. All FLAG-based Stx17 constructs were constructed as described previously[15]. 3 × -FLAG-Atg14L and 3 × -FLAG-DFCP1 (template for the FLAG-DFCP1 construct) were generous gifts from Dr Noboru Mizushima (University of Tokyo).

**Cell culture and transfection.** HeLa cells were obtained from the RIKEN Bioresource Center (RCB0007) and found to be free from mycoplasma contamination. HEK293 cells stably expressing FcγRII (HEK293-FcγRII cell) were maintained as described previously[11]. HeLa cells stably expressing FcγRII (HeLa-FcγRII cell) were established as described previously[11], and grown in α-MEM supplemented with

50 IU ml$^{-1}$ penicillin, 50 μg ml$^{-1}$ streptomycin, 10% fetal calf serum, 2 mM L-glutamine and 400 μg ml$^{-1}$ hygromycin (Wako Chemicals). HeLa cells stably expressing wild-type FLAG-Stx17 or the K254C mutant were maintained as described previously[15]. THP1 monocytes were obtained from the RIKEN Bioresource Center (RCB1189) and grown in RPMI-1640 medium supplemented with 50 IU ml$^{-1}$ penicillin, 50 μg ml$^{-1}$ streptomycin and 10% fetal calf serum. THP1 monocytes were induced to differentiate into macrophages by treatment with 10 ng ml$^{-1}$ phorbol 12-myristate 13-acetate (Wako Chemicals) for 48 h. Transfection was carried out using Lipofectamine 2000 (Thermo Fisher Scientific) according to the manufacturer's protocol.

**Chemicals and antibodies.** Chemicals were obtained from the following sources: Hoechst 33342 (Thermo Fisher Scientific), PMSF (Wako chemicals), MG132 (Calbiochem), TRAIL (Merck Millipore), STS (Wako chemicals) and Percoll

(Sigma-Aldrich). The following antibodies were obtained from Sigma-Aldrich: α-tubulin (No. T6074; 1:2,000 dilution for IB), polyclonal FLAG (No. F7425; 1:3,000 dilution for IB and 1:200 dilution for immunofluorescence (IF)) and monoclonal FLAG (No. F3165; 1:200 dilution for IF). The following antibodies were obtained from BD Bioscience Pharmingen: cytochrome c (No. 556432; 1:200 dilution for IF), CNX (No. 610523; 1:1,000 dilution for IB), Drp1/Dlp1 (No. 611112; 1:40 dilution for IF) and Tom20 (No. 612278; 1:500 dilution for IB and 1:200 dilution for IF). The following antibodies were from MBL: polyclonal LC3 (No. PM036; 1:1,000 dilution for IB), monoclonal LC3 (No. M152-3; 1:150 dilution for IF), Atg14L (No. M184-3; 1:500 dilution for IB) and beclin-1 (No. PD017; 1:500 dilution for IB). The following antibodies were from the indicated sources: GFP (Thermo Fisher Scientific, No A6455; 1:3,000 dilution for IB), MBP (New England Biolabs, No. E8030; 1:2,000 dilution for IB), cleaved caspase-3 (Cell Signaling, No. 9661; 1:100 dilution for IF) and Bax (Santa Cruz Biotechnology, No. sc-493; 1:100 dilution for IF). A rabbit serum against *Legionella*[11] was used at 1:3,000 dilution for opsonization and IF, respectively. A mouse serum against *Legionella* was prepared by immunizing mice with heat-killed *Legionella* and used at 1:2,000 and 1:3,000 dilution for opsonization and IF, respectively. Antisera against LidA and Lpg1137 were prepared by immunizing rabbits with GST-LidA and MBP-Lpg1137, respectively, and used at 1:1,000 and 1:200 dilution for IB, respectively. Other rabbit antibodies used were anti-Stx17 (ref. 15) (1:1,000 dilution for IB and 1:100 dilution for IF), anti-Stx18 (1:500 dilution for IB)[46] and Sec22b (1:500 dilution for IB)[46]. Alexa Fluor 488 and 594 goat anti-mouse and -rabbit antibodies (Nos. A-11001, A-11005, A-11008 and A-11012; 1:100 dilution, respectively) were obtained from Thermo Fisher Scientific. Horseradish peroxidase-conjugated anti-rabbit and -mouse antibodies (No. 170-6515 (1:3,000 dilution) and 170-6516 (1:3,000 dilution), respectively) were purchased from Bio-Rad.

**Immunofluorescence.** For IF microscopy, cells were fixed with 4% paraformaldehyde for 20 min at room temperature and then observed under an Olympus Fluoview 300 laser scanning microscope or BX53 microscope with a DP53 CCD camera. In the *Legionella* infection experiments, extracellular bacteria were detected using rabbit or mouse anti-*Legionella* antiserum and then stained. After staining, cells were permeabilized and further stained for host proteins and intracellular bacteria.

**Subcellular fractionation.** Cells expressing GFP or GFP-Lpg1137 were homogenized in buffer (30 mM Tris-HCl (pH 7.4), 225 mM mannitol and 75 mM sucrose) by passage 15 times through a 26 G needle. The homogenates were centrifuged at 600 g for 5 min to obtain a postnuclear supernatant. The postnuclear supernatant was centrifuged at 7,000 g for 10 min to pellet the crude mitochondrial fraction. The supernatant was centrifuged at 100,000 g for 30 min to separate the cytosol and microsomal fractions. The crude mitochondrial fraction was suspended in buffer (5 mM HEPES (pH 7.4), 250 mM mannitol and 0.5 mM EGTA), loaded onto a 30% Percoll in buffer (25 mM HEPES (pH 7.4), 225 mM mannitol and 1 mM EGTA) and centrifuged at 95,000 g for 30 min. After centrifugation, the MAM and mitochondria fractions were collected. The protein concentration of each fraction was measured, and equal amounts of proteins were subjected to SDS–polyacrylamide gel electrophoresis and then analysed by IB.

**Protein purification and *in vitro* degradation assay.** His$_6$-tagged Stx17 full-length was expressed in *E. coli* (BL21 codon-plus RP strain; Agilent Technologies) and solubilized in buffer containing 25 mM HEPES-KOH (pH 7.4), 500 mM NaCl, 1 mM MgCl$_2$, 1 mM dithiothreitol, 1 mM PMSF and 1% of n-dodecyl-*N*,*N*-dimethylamine-N-oxide (Wako Chemicals). His$_6$-Stx17 full-length was purified using Ni-NTA beads (Qiagen). MBP protein, MBP-tagged Lpg1137 wild-type and Lpg1137 S68A were expressed in *E. coli* (BL21 codon plus RP strain) and then solubilized in buffer containing 25 mM HEPES-KOH (pH 7.4), 500 mM NaCl, 1 mM MgCl$_2$ 1 mM dithiothreitol and 1% of Triton X-100. The MBP proteins were purified using amylose resin (New England Biolabs). For *in vitro* degradation assay, 0.2 μg of Stx17 full-length was mixed with an equal amount of an MBP protein in reaction buffer containing 25 mM HEPES-KOH (pH 7.4), 50 mM NaCl and 5 mM MgCl$_2$, followed by incubation at 37 °C for 0, 30 or 60 min. The reaction was stopped by the addition of SDS sample buffer.

**Immunoprecipitation.** HEK293-FcγRII cells expressing FLAG-tagged proteins were lysed in lysis buffer containing 20 mM HEPES-KOH (pH 7.2), 150 mM KCl, 2 mM EDTA, 1 mM dithiothreitol, 1 μg ml$^{-1}$ leupeptin, 1 μM pepstatin A, 2 μg ml$^{-1}$ aprotinin, 1 mM PMSF and 1% Triton X-100. After centrifugation, the supernatants were immunoprecipitated with anti-FLAG M2 affinity beads (Sigma-Aldrich). The bound proteins were eluted with SDS sample buffer and then analysed by IB.

**RNA interference.** The siRNAs used were described previously[15] except for Atg14L (5′-UUUGCGUUCAGUUUCCUCACUGCGC-3′). siRNAs were purchased from Japan Bio Services (Asaka, Japan). HeLa-FcγRII cells were grown on 35-mm dishes, and siRNAs were transfected at a final concentration of 200 nM

using Oligofectamine (Thermo Fisher Scientific) according to the manufacturer's protocol.

**Proximity ligation assays.** PLA reagents were obtained from Sigma-Aldrich (catalogue numbers DUO92002, DUO92004 and DUO92008), and the assay was performed according to the manufacturer's protocol.

**Induction of autophagy.** To induce autophagy, cells were washed with PBS three times and then incubated with Earle's balanced salt solution (EBSS) for 2 h. To block autophagic flux, 0.1 μM bafilomycin A1 (Calbiochem) was included in EBSS.

**Quantification of LC3-positive structures.** As LC3-positive structures varied in size and large LC3-positive structures often aggregated in starved cells, we measured the LC3 immunostaining intensity, not the number of LC3-positive dots, of each cell and compared. In this measurement, the intensity of LC3 signal was scored with the same threshold that eliminated almost cytosolic background of the LC3 signal.

**Intracellular growth assay.** Bone marrow macrophages were collected from C57BL/6 NLRC4 − / − mice[47] and plated in differentiation medium (RPMI supplemented with 20% fetal calf serum and 15% L929 supernatant). Differentiated bone marrow-derived macrophages were seeded at 2 × 10$^5$ per well into 24-well tissue culture-treated plates one day before infection in replating medium (RPMI supplemented with 10% fetal calf serum and 7.5% L929 supernatant). Two-day patches of *L. pneumophila* strains were resuspended in sterile water and used to infect macrophages at an MOI of 1 in triplicates. Plates were centrifuged at 250 g to facilitate bacterial attachment. One hour postinfection, cells were washed three times in sterile PBS and 1 ml of replating medium was added back to the cells. Bacteria were liberated by hypotonic lysis at 1, 24, 48 and 72 h postinfection, and colony-forming units were enumerated on CYE agar after 4 days of growth at 37 °C.

**Quantification and statistics.** To quantify nuclear condensation, cleavage of caspase-3 and cytochrome c release, at least 50 cells were counted in each experiment. Quantification of the fluorescence intensity of Bax, FLAG-DFCP1 and 3 × -FLAG-Atg14L was performed using the ImageJ software. To quantify the number of PLA dots, 30 cells were scored in each experiment. The results were averaged, expressed, unless otherwise stated, as the mean with standard deviation and then analysed by means of a paired Student's t-test. The P values are indicated by asterisks in the figures with the following notations: *P < 0.01, **P < 0.001 and ***P < 0.0001.

**Data availability.** The data supporting the findings of this study are available within the article and its Supplementary Information files, or from the corresponding authors on request.

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

## Acknowledgements

We thank Ms Sakiko Matsumoto, Dr James C. Havey and Dr Craig R. Roy for their help in this study. We thank Dr R.R. Isberg and Dr N. Mizushima for the generous gifts of *Legionella* mutants and plasmids, respectively. This work was supported in part by Grants-in-Aid for Scientific Research (Nos 25291029 and 26650066 to M.T. and Nos 26111520 and 26713016, and 16H01206 to K.A.) and the MEXT-Supported Program for the Strategic Research Foundation at Private Universities (to M.T. and K.A.) from the Ministry of Education, Culture, Sports, Science and Technology of Japan; the Uehara Memorial Foundation (to K.A.); and the Sumitomo Foundation (to K.A.).

## Author contributions

K.A. and M.T. conceived the project. K.A. and Y.M. performed experiments. S.R.S constructed *Legionella* mutants. H.I. and Y.W. supported experiments.

## Additional information

**Competing interests:** The authors declare no competing financial interests.

