## [Peer Review File · Nature Communications]

Reviewers' comments:

Reviewer #1 (Remarks to the Author):

The manuscript by Arasaki et al shows that the Lpg1137 effector of Legionella is a serine protease that specifically cleaves syntaxin 17, leading to strong effects on autophagy and cell death. This is an interesting and original observation, and the data supporting the conclusion are convincing and clear for the most part. There are however a couple of issues that need to be addressed:

Major: It was not clear what effect Lpg1137 has on the normal infectivity cycle of Legionella. Since the authors have the inactive mutant they must address this question, preferably by experiments. They have done a good job showing the effects of Lpg137 on the host cells, but need to also show effects (if any) on the pathogen.

Minor: The data showing effects of Lpg1137 on autophagy, and especially using LC3 blots (Suppl Fig 3b), indicate a strong effect of Stx17 reduction on the total levels of LC3. This appears to be at variance with the siRNA data from Itakura et al, which showed an effect on flux but not on LC3 levels. It almost appears that Lpg1137 causes LC3 degradation as well. Also, it would be better to do this experiment +/- Bafilomycin in order to properly measure flux.

Minor: Can the authors try to provide a higher resolution image of the Lpg1137 localization (e.g. Suppl Fig 1) to see if there is some ER/MAM signal?

Minor: Can the authors validate their Stx17 antibody used for iffl (e.g. Fig 1 a, c) especially since it is one of the few antibodies that works for this application? I would like to see an siRNA experiment for Stx17 followed by iffl.

Minor: The Drp1 antibody listed in Materials and Methods is actually against Dlp1. Can the authors clarify what it is they used?

Reviewer #2 (Remarks to the Author):

The article "Legionella effector Lpg1137 shuts down ER-mitochondria communication through cleavage of syntaxin 17" by Arasaki and colleagues reports the identification of an effector of Legionella pneumophila, that encodes a serine protease that cleaves syntaxin 17 in the host cell. The authors show that the interaction of Lpg1137 with Stx17 leads to inhibition of autophagy and staurosporine-induced apoptosis.

The identification of this new effector is interesting, however, the here presented work does not demonstrate the mechanism claimed in the abstract of Lpg1137 disrupting DRP1-Stx17 and ATG14L-Stx17 interactions, with consequences on autophagy or apoptosis, in particular not during infection as the experiments proposed for explaining the mechanism are done with transfection and not infection. There is no proof of DRP1-Stx17 interaction during wt, mutant or DotA infection, or any proof of WT-induced disruption of these interactions. As this mechanism is clearly suggested, authors should address this point by quantifying the interaction between DRP1/ATG14L and Stx17 during WT, DotA and mutant infection, as well as during overexpression of wt -Lpg1137 compared to a protein mutated in key residues.

Page 2, last line, please explain what "the isogenic mutant with large chromosomal deletions is".... a non Legionella reader has no idea

Page 3, first line and Figure 2... the same, please explain what the delta 2 mutant is.

Page 3, lines 7-9 it reads... To confirm that Lpg1137 is responsible for Stx17 cleavage, we constructed a transposon mutant strain (Lpg1137 TM)... It is unclear to that reviewer why you constructed a transposon mutant and not a clean deletion mutant? This should be done.

Page 3, second paragraph and figure 3... this experiments do not clearly demonstrate that Lpg1137 targets MAM/mitochondria; You propose as markers calnexin (ER marker) and Tom20 (mitochondria marker), but this result does not exclude that you recognize a mixture of mitochondria and ER but not MAMs. Thus a specific MAM marker (e.g. ACS4/FACL4, PEMT2, PSS1/2 or DGAT2) needs to be added to this blot. I suggest you do a fractionation that shows, PNS, Cytosol, MS, Mitochondria, ER and MAMs. This would clearly establish whether the protein is indeed found in MAMs.

For this reviewer it is not clear how the autophagy part of this paper was analysed. E.g. Supplementary Figure 3a it is indicated that fluoresce intensity was measured. However, as there is always background with antibody staining, how the authors did account for background; Normally the number of LC3 puncta is evaluated; This must be better described or redone and puncta counts should be presented.

The authors use RavZ as another effector that interferes with autophagy to demonstrate that STx17 participates in an early stage of autophagy. However, there was recently a publication on another effector, LpSpl (Rolando et al., 2016, PNAS) that also interferes with autophagy. This effector should also be analysed in this context and in particular discussed in the article. How all these effectors interplay to regulate autophagy to the pathogens advantage?

The effector Lpg1137 described here is specifically present in strain *L. pneumophila* Philadelphia, but absent from the other *L. pneumophila* strains commonly studied. This is intriguing. The authors should discuss this finding and undertake experiments to see whether other strains are also able to cleave STx17 or if this is a specificity of the Philadelphia strain used in this study.

To Reviewer #1

Major Comment 1 ---- It was not clear what effect Lpg1137 has on the normal infectivity cycle of Legionella. Since the authors have the inactive mutant they must address this question, preferably by experiments. They have done a good job showing the effects of Lpg137 on the host cells, but need to also show effects (if any) on the pathogen.

Reply ---- Thank you very much for this valuable comment. We have performed an intracellular growth assay. Fig. 2d shows that the transposon mutant *Lpg1137TM* and the newly constructed deletion mutant Δ *Lpg1137* grew normally, suggesting that loss of Lpg1137 does not significantly affect pathogen viability.

Minor Comment 1 ---- The data showing effects of Lpg1137 on autophagy, and especially using LC3 blots (Suppl Fig 3b), indicate a strong effect of Stx17 reduction on the total levels of LC3. This appears to be at variance with the siRNA data from Itakura et al, which showed an effect on flux but not on LC3 levels. It almost appears that Lpg1137 causes LC3 degradation as well. Also, it would be better to do this experiment +/- Bafilomycin in order to properly measure flux.

Reply ---- Following the suggestion by the reviewer, we have performed a flux assay in the presence of bafilomycin A1, and the formation of LC3-II was analyzed by immunoblotting. As show in Fig. 6b, the accumulation of LC3-II was significantly blocked in GFP-Lpg1137-expressing cells in the presence of bafilomycin A1 as well as its absence, consistent with our previous data demonstrating that siRNA-mediated Stx17 knockdown blocked the accumulation of LC3-II upon starvation (Arasaki et al. (2015) Dev Cell 32, 304).

Minor Comment 2 ---- Can the authors try to provide a higher resolution image of the Lpg1137 localization (e.g. Suppl Fig 1) to see if there is some ER/MAM signal?

Reply ---- Because of the presence of Lpg1137 in the cytosol, it is difficult to detect its signal in the MAM at the level of immunofluorescence microscopy. Instead, we have performed subcellular fractionation. The data showed that GFP-Lpg1137 was localized in the MAM fraction, where the MAM marker FAACL4 was present (Fig. 3a, right).

Minor Comment 3 ---- Can the authors validate their Stx17 antibody used for iffl (e.g. Fig 1 a, c) especially since it is one of the few antibodies that works for this application? I would like to see an siRNA experiment for Stx17 followed by iffl.

Reply ---- As described in Methods, we used a “home-made” anti-Stx17 antibody. Its

specificity in immunofluorescence microscopic analysis was already verified (Arasaki et al. (2015) Dev Cell 32, 304, Fig. 2B).

Minor Comment 4 ---- The Drp1 antibody listed in Materials and Methods is actually against Dlp1. Can the authors clarify what it is they used?

Reply ---- Dlp1 is another name for Drp1. In the “chemicals and antibodies” in Methods, we have changed “Drp1” to “Drp1/Dlp1”.

To Reviewer #2

Comment 1 ---- The identification of this new effector is interesting, however, the here presented work does not demonstrate the mechanism claimed in the abstract of Lpg1137 disrupting DRP1-Stx17 and ATG14L-Stx17 interactions, with consequences on autophagy or apoptosis, in particular not during infection as the experiments proposed for explaining the mechanism are done with transfection and not infection. There is no proof of DRP1-Stx17 interaction during wt, mutant or DotA infection, or any proof of WT-induced disruption of these interactions. As this mechanism is clearly suggested, authors should address this point by quantifying the interaction between DRP1/ATG14L and Stx17 during WT, DotA and mutant infection, as well as during overexpression of wt -Lpg1137 compared to a protein mutated in key residues.

Reply ---- Thank you very much for a very important and valuable comment. We have performed a proximity ligation assay (PLA) to measure the proximity of Stx17 to 3x-FLAG-Atg14L/Drp1 during wild-type, *dotA*, or *Lpg1137 TM* infection. The results showed that the proximity of Stx17-3x-FLAG-Atg14L and Stx17-Drp1 was drastically abolished during WT *Legionella* infection, but not infection with either mutant (Fig. 5a and Supplementary Fig. 2a). Expression of GFP-Lpg1137 wild-type, but not the S68A mutant, also abolished the PLA signals (Fig. 5b and Supplementary Fig. 2b).

Comment 2 ---- Page 2, last line, please explain what “he isogenic mutant with large chromosomal deletions is ---- ”, a non Legionella reader has no idea. Page 3, first line and Figure 2 ----, the same, please explain what the delta 2 mutant is.

Reply ---- The following explanation has been added to page 4, second paragraph; “*L. pneumophila* chromosome has a modular architecture, and five genomic islands (islands 2, 3, 4, 6 and 7) are devoid of genes for growth in rich media, but exhibit a high concentration of effectors that facilitate infection and proliferation in host cells²².”

Comment 3 ---- Page 3, lines 7-9 it reads ---- To confirm that Lpg1137 is responsible for Stx17 cleavage, we constructed a transposon mutant strain (Lpg1137 TM)----. It is

unclear to that reviewer why you constructed a transposon mutant and not a clean deletion mutant? This should be done.

Reply ---- Following the suggestion of the reviewer, we have made a clean Lpg1137 deletion strain (Δ Lpg1137), and demonstrated that both this mutant and the transposon mutant lacked Lpg1137 protein (Fig. 2b) and the activity to degrade Stx17 (Fig. 2c).

Comment 4 ----Page 3, second paragraph and figure 3 --- this experiments do not clearly demonstrate that Lpg1137 targets MAM/mitochondria; You propose as markers calnexin (ER marker) and Tom20 (mitochondria marker), but this result does not exclude that you recognize a mixture of mitochondria and ER but not MAMs. Thus a specific MAM marker (e.g. ACS4/FACL4, PEMT2, PSS1/2 or DGAT2) needs to be added to this blot. I suggest you do a fractionation that shows, PNS, Cytosol, MS, Mitochondria, ER and MAMs. This would clearly establish whether the protein is indeed found in MAMs.

Reply ---- This comment is related to a comment of Reviewer #1. To address this, we have performed subcellular fractionation to obtain the PNS, Cytosol, MS, Mitochondria, ER and MAM fractions, and analyzed by immunoblotting with antibodies including one against the MAM marker FACL4. The data showed the presence of GFP-Lpg1137 in the MAM fraction, in addition to in other fractions (Fig. 3a).

Comment 5 ---- For this reviewer it is not clear how the autophagy part of this paper was analysed. E.g. Supplementary Figure 3a it is indicated that fluoresce intensity was measured. However, as there is always background with antibody staining, how the authors did account for background; Normally the number of LC3 puncta is evaluated; This must be better described or redone and puncta counts should be presented.

Reply ---- As pointed out by the reviewer, the number of GFP (or other tag) -LC3 puncta has been evaluated in many studies. Although expressed LC3 gave discrete puncta in starved cells, endogenous LC3 puncta varied in size and was frequently observed as large-aggregated structures, which made it difficult to quantify. We therefore chose LC3 staining intensity as an index of autophagosome formation. In this measurement, we scored the intensity of LC3 signal with the same threshold that eliminated the cytosolic background. The method to quantify LC3-positive structures is stated in Methods.

Comment 6 ---- The authors use RavZ as another effector that interferes with autophagy to demonstrate that STx17 participates in an early stage of autophagy. However, there was recently a publication on another effector, LpSpl (Rolando et al., 2016, PNAS) that

also interferes with autophagy. This effector should also be analysed in this context and in particular discussed in the article. How all these effectors interplay to regulate autophagy to the pathogens advantage?

Reply ---- Thank you very much for informing this important paper, which we obviously missed. Because RavZ cleaves the covalent bond between LC3 and phosphatidylethanolamine, thereby preventing autophagosome formation, we used this to examine whether Stx17 localizes to the omegasome structure where isolation membranes/autophagosomes are formed. The experiment suggested by the reviewer is an important one that should be performed in future study. Therefore, we have added at the end of discussion section the following sentences; “In the course of this study, it was reported that the *Legionella* effector sphingosine-1 phosphate lyase (LpSpl) interferes with autophagy by affecting host sphingolipid metabolism⁴⁰. Future work is required to reveal the interplay among multiple effectors (RavZ, Stx17 and LpSpl) to suppress autophagy, as well as the strategic advantage for the presence of the multiple autophagy suppressors in infection and intracellular growth of *Legionella*.”

Comment 7 ---- The effector Lpg1137 described here is specifically present in strain L. pneumophila Philadelphia, but absent from the other L. pneumophila strains commonly studied. This is intriguing. The authors should discuss this finding and undertake experiments to see whether other strains are also able to cleave STx17 or if this is a specificity of the Philadelphia strain used in this study.

Reply ---- A recent paper by Borges *et al.* (Sci. Rep. 6:26261 (2016); Supplementary Table 2) showed that Lpg1137 is present in many *L. pneumophila* strains including *L. pneumophila* Paris, Lens, Corby, 2300/99 Alcoy, 130b and PtVFX/2014, not peculiar to *L. pneumophila* Philadelphia.

REVIEWERS' COMMENTS:

Reviewer #1 (Remarks to the Author):

I am satisfied with the modifications in the resubmitted manuscript.

Reviewer #2 (Remarks to the Author):

The revised version of this manuscript is much improved and the authors have answered all my comments and critics adequately. I thus have no more comments.